# Hybrid Cramér-Rao Bound for Quantum Bayes Point Estimation with Nuisance Parameters

**DOI:** 10.3390/e27121184

**Published:** 2025-11-21

**Authors:** Jianchao Zhang, Jun Suzuki

**Affiliations:** 1Graduate School of Informatics and Engineering, The University of Electro-Communications, 1-5-1 Chofugaoka, Chofu-shi, Tokyo 182-8585, Japan; junsuzuki@uec.ac.jp; 2Institute for Advanced Science, The University of Electro-Communications, 1-5-1 Chofugaoka, Chofu-shi, Tokyo 182-8585, Japan

**Keywords:** quantum parameter estimation, nuisance parameters, quantum Fisher information, quantum Cramér–Rao bound

## Abstract

We develop a hybrid framework for quantum parameter estimation in the presence of nuisance parameters. In this scheme, the parameters of interest are treated as fixed non-random parameters while nuisance parameters are integrated out with respect to a prior (random parameters). Within this setting, we introduce the hybrid partial quantum Fisher information matrix (hpQFIM), defined by prior-averaging the nuisance block of the QFIM and taking a Schur complement, and derive a corresponding Cramér–Rao-type lower bound on the hybrid risk. We establish the structural properties of the hpQFIM, including inequalities that bracket it between computationally tractable approximations, as well as limiting behaviors under extreme priors. Operationally, the hybrid approach improves over pure point estimation since the optimal measurement for the parameters of interest depends only on the prior distribution of the nuisance, rather than on its unknown value. We illustrate the framework with analytically solvable qubit models and numerical examples, clarifying how partial prior information on nuisance variables can be systematically exploited in quantum metrology.

## 1. Introduction

Quantum metrology and quantum sensing have matured into rigorous frameworks for quantum-limited precision measurement, with rapid theoretical and experimental progress in recent years [1,2,3,4,5,6]. In many such tasks, the parameter vector naturally separates into parameters of interest, which encode the physical quantity we ultimately care about, and nuisance parameters, which affect the data but are not themselves the target [7,8]. Typical nuisances include optical loss and detector inefficiency in interferometry [9], unknown phase or polarization offsets due to misalignment, dephasing and amplitude-damping rates in spectroscopy [1], background counts in imaging [10], or slow drifts in local oscillators for frequency standards [11]. Treating interest and nuisance on equal footing can blur the operational goal and can also reduce statistical efficiency [8]: measurement settings that are ideal for learning the nuisance may be suboptimal for the scientific quantity of interest [12].

Quantum estimation provides the decision-theoretic backbone for metrology and sensing by linking experimental design (choice of measurement) to achievable precision limits. Quantum parameter estimation admits both frequentist and Bayesian formulations [13,14]. In point (frequentist) formulations, locally optimal measurements often depend on the unknown true parameter; in multi-parameter models, incompatibility between observables can prevent simultaneous attainment of single-parameter limits [15]. In fully Bayesian formulations, performance is optimized on average with respect to a prior, which improves robustness and ease of implementation but may reduce local efficiency when the prior is diffuse or misspecified [16]. In practice, such as in atomic clocks [17], magnetometry [18], optical phase tracking [19], and nanoscale imaging [20], we often have partial prior information about nuisance parameters from routine characterization, while the scientific parameters of interest still demand local, high-resolution treatment [8]. This operational asymmetry motivates a hybrid approach.

### 1.1. Contributions of This Paper

Framework and risk. We formalize a hybrid estimation framework that treats parameters of interest as fixed non-random parameters while incorporating nuisance parameters through a prior distribution, i.e., random parameters obeying the distribution. We introduce a hybrid mean squared error (MSE) and hybrid risk as the objective to minimize (Definition 1).Hybrid CR-type lower bound. We prove a Cramér–Rao-type (CR-type) inequality in the hybrid setting, identifying the hybrid partial quantum Fisher information matrix (hpQFIM) as a fundamental lower bound on the hybrid risk for the interest parameters under admissible measurements and estimators (Theorem 1).Two-sided approximations and ordering relations. We establish computable upper and lower approximations for the hpQFIM (Theorem 2).

**Remark** **1.**
*In this framework, only the parameter of interest is estimated pointwisely. The nuisance parameter is not estimated, as it is not of interest.*


### 1.2. Short Summary of Point Estimation and Bayesian Estimation

#### 1.2.1. Point Estimation in Quantum Models

In point estimation, one fixes an unknown parameter value and seeks measurements and estimators that are locally efficient around that point. Classical CR-type guarantees relate the achievable MSE to information carried by the measurement outcomes, and in quantum settings, the choice of measurement becomes part of the optimization itself [21,22]. In multi-parameter models, jointly optimal measurements can be hindered by incompatibility among observables [23]; Holevo-type criteria capture the best trade-offs permitted by quantum mechanics [24,25]. A practical limitation is that locally optimal measurements typically depend on the unknown parameter, so adaptive or two-stage strategies are often used to first localize and then refine [25,26,27].

#### 1.2.2. Bayesian Estimation and Prior-Averaged Optimality

Bayesian estimation evaluates performance on average with respect to a prior over parameters. The optimal measurement-estimator pair minimizes the Bayes risk defined by this prior, and fundamental lower bounds—such as van Trees-type inequalities and their quantum analogues—link Bayes risk to prior-averaged information quantities [16,28]. In quantum settings, several quantum versions of these classical bounds have been proposed [29,30,31]. Recent Bayesian logarithmic-derivative (LD)-type bounds provide convenient and often tighter computable benchmarks in finite-copy regimes [32]. In practice, the optimal Bayesian measurement depends on the prior rather than the unknown true value; this reduces design complexity when only distributional knowledge is available or when adaptive localization is expensive.

#### 1.2.3. Motivation for a Hybrid Framework

Point estimation excels at local precision but can require rapid localization and may face incompatibility in the multi-parameter regime. Full Bayesian estimation is robust and measurement-friendly, yet local sharpness can be diluted under diffuse or misspecified priors. Many metrological scenarios lie between these extremes: we often possess actionable prior information about nuisance components while still aiming for the best local performance on the parameters of interest. This motivates a hybrid approach that uses prior averaging for nuisance parameters to gain robustness and implementability, while preserving pointwise efficiency for the parameters of interest. This perspective resonates with the hybrid CR lower bound in classical signal processing [33,34]. In this work, we extend these ideas to quantum estimation; formal definitions and bounds shall be presented in the paper.

The remainder of this paper is organized as follows. In Section 2, we formulate the hybrid estimation framework, define the hpQFIM, and derive the associated hybrid CR-type lower bound together with computable inequalities. In Section 3, we present numerical case studies on noisy qubit models to illustrate the behavior of the proposed hybrid bound under different nuisance structures. In Section 4, we analyze an analytically solvable qubit example where directional parameters are estimated in the presence of a radial nuisance, highlighting how the hybrid formulation connects to the state. We also report a numerical simulation for finite-samples. Section 5 concludes the paper and discusses open directions. Detailed proofs of the main theorems are provided in Appendix A and Appendix B.

## 2. Hybrid Framework

In this section, we present the hybrid framework and two main theorems. The proofs of the theorems are written in the appendix.

### 2.1. Setting and Notation

The parametric model is $\left\{\rho_{\theta_{I},\theta_{N}}|\theta_{I}\in \Theta_{I}\subset R^{d_{I}},\;\theta_{N}\in \Theta_{N}\subset R^{d_{N}}\right\}$ with the state $\rho_{\theta_{I},\theta_{N}}$ on a finite dimensional Hilbert space H. Let the parameter vector be partitioned as $\theta =(\theta_{I},\theta_{N})$ such that(1)$$\begin{array}{c} \theta =\left(\theta_{1},\theta_{2},\cdots ,\theta_{d_{I}},\theta_{d_{I}+1},\cdots ,\theta_{d_{I}+d_{N}}\right)\in \Theta_{I}\times \Theta_{N}\subset R^{d_{I}+d_{N}}, \end{array}$$
where $d_{I}$ and $d_{N}$ represent the numbers of parameters of interest and nuisance parameters. Thus we use $\theta_{I}$ and $\theta_{N}$ to represent the vectors of corresponding parameters. The positive operator-valued measure (POVM) is $\Pi =\{\Pi_{x}|x\in X\}$ with outcome $x∼p\left(x\;|\;\theta_{I},\theta_{N}\right)=\mathrm{tr}\left[\rho_{\theta_{I},\theta_{N}}\Pi_{x}\right]$ such that $\Pi_{x}⪰0$ and(2)$$\begin{array}{c} \int_{X}\Pi_{x}\;dx=I, \end{array}$$
where the integral is taken over X. In the purely discrete case, the integral is understood as a sum, i.e., $\sum_{x\in X}\Pi_{x}=I$. We use the integral notation for uniformity and all statements apply equally to discrete outcome spaces.

To estimate the parameter of interest $\theta_{I}$, we construct an estimator ${\hat{\theta }}_{I}$. In the present framework, the nuisance parameter $\theta_{N}$ would not be estimated since it is not of interest. This is one of the advantages of this hybrid estimation framework. We assume that the estimator ${\hat{\theta }}_{I}$ satisfies the locally unbiased condition as follows.

#### 2.1.1. Locally Unbiased and Boundary Conditions

The estimator ${\hat{\theta }}_{I}:X\mapsto \Theta_{I}$ is locally unbiased at $\theta_{I}$ for any $\theta_{N}$, which means that(3)$$\begin{array}{cc} E_{x|\theta_{I},\theta_{N}}\left[{\hat{\theta }}_{I,i}\left(x\right)\right] & =\theta_{I,i}\;\;\mathrm{for}\;\mathrm{all}\;i,\\ \frac{\partial }{\partial \theta_{I,i}}E_{x|\theta_{I},\theta_{N}}\left[{\hat{\theta }}_{I,j}\left(x\right)\right] & =\delta_{i,j}\;\;\mathrm{for}\;\mathrm{all}\;i,j, \end{array}$$
where $\theta_{I,i}$ denotes the *i*-th parameter of interest and the expectation is(4)$$\begin{array}{cc} E_{x|\theta_{I},\theta_{N}}\left[f\left(x\right)\right] & :=\int_{X}f\left(x\right)p\left(x|\theta_{I},\theta_{N}\right)dx. \end{array}$$

In this research, the random variable is denoted by lowercase *x*.

#### 2.1.2. Boundary Conditions

We will use the following minimal regularity/boundary assumptions:Interchange of derivative and integral.For any measurable *f*, differentiation w.r.t. $\theta_{I}$ can be passed under the integral sign:$$\partial_{\theta_{I}}\;\int_{X}f\left(x\right)\;p\left(x|\theta_{I},\theta_{N}\right)\;dx=\int_{X}f\left(x\right)\;\partial_{\theta_{I}}p\left(x|\theta_{I},\theta_{N}\right)\;dx,$$
and, after marginalizing $\theta_{N}$ with a $\theta_{I}$-independent prior $\pi_{N}\left(\theta_{N}\right)$,$$\partial_{\theta_{I}}p\left(x|\theta_{I}\right)=\int_{\Theta_{N}}\partial_{\theta_{I}}p\left(x|\theta_{I},\theta_{N}\right)\;\pi_{N}\left(\theta_{N}\right)\;d\theta_{N}.$$Prior tail for integration by parts.Whenever an integration by parts in a parameter $\theta_{N}$ with prior density $\pi (\theta_{N})$ is used, assume that $\pi (\theta_{N})$ decays fast enough at the boundary.

These assumptions are used for Theorem 1.

#### 2.1.3. The Main Objective

The original main aim of quantum estimation is finding the following quantity:(5)$$\begin{array}{cc} \; & \min_{\Pi ,{\hat{\theta }}_{I}}V_{\theta_{I},\pi }\left(\Pi ,{\hat{\theta }}_{I}\right), \end{array}$$(6)$$\begin{array}{cc} \; & s.t.\;{\hat{\theta }}_{I}\;\mathrm{is}\;\mathrm{locally}\;\mathrm{unbiased}\;\mathrm{at}\;\theta_{I}\;\mathrm{for}\;\mathrm{any}\;\theta_{N}. \end{array}$$

This minimization is taken over $\left(\Pi ,{\hat{\theta }}_{I}\right)$, which is called the quantum decision. However, this minimization is not always possible because it is a matrix. Thus, the main objective is to minimize the scalar hybrid risk for a weight matrix, W≻0 on parameters of interest $R^{d_{I}}$. The hybrid risk is defined as follows.

**Definition** **1**(Hybrid risk)**.**(7)$$\begin{array}{c} R_{\theta_{I},\pi }\left(\Pi ,{\hat{\theta }}_{I}|W\right):=\mathrm{Tr}\left[W\cdot V_{\theta_{I},\pi }\left(\Pi ,{\hat{\theta }}_{I}\right)\right]. \end{array}$$

The prior $\pi (\theta_{N})$ is assumed to be continuously differentiable twice in $\theta_{N}$ and decays sufficiently fast at the boundary (or at infinity) so that all boundary terms vanish in first-order integration by parts.

#### 2.1.4. Scope

Throughout this paper, we perform frequentist point estimation for the parameters of interest $\theta_{I}$, while marginalizing nuisance parameters $\theta_{N}$ under a prior $\pi_{N}\left(\theta_{N}\right)$ that is independent of $\theta_{I}$. The analyzes of biased estimators for $\theta_{I}$ are beyond the present scope.

### 2.2. Quantum Information Blocks and the hpQFIM

Let *J* denote the symmetric logarithmic derivative (SLD) quantum Fisher information matrix (QFIM) and write its block form:(8)$$\begin{array}{c} J\left(\theta_{I},\theta_{N}\right)=\left(\begin{array}{cc} J_{II}\left(\theta_{I},\theta_{N}\right) & J_{IN}\left(\theta_{I},\theta_{N}\right)\\ J_{NI}\left(\theta_{I},\theta_{N}\right) & J_{NN}\left(\theta_{I},\theta_{N}\right) \end{array}\right)\in R^{\left(d_{I}+d_{N}\right)\times \left(d_{I}+d_{N}\right)}. \end{array}$$

In this research, the SLD QFIM is involved since this is the most widely used quantum score operator due to its symmetric and Hermitian properties [21]. However, the result is free to extend to the right logarithmic derivative (RLD) QFIM [35]. In what follows, we omit the explicit $\left(\theta_{I},\theta_{N}\right)$ dependence in QFIM blocks when clear from context. We use the partial QFIM for the parameters of interest, defined as the Schur complement of the nuisance block:(9)$$\begin{array}{c} J_{I|N}\;:=\;J_{II}\;-\;J_{IN}\;J_{NN}^{-1}\;J_{NI}, \end{array}$$
which captures the information on $\theta_{I}$ after optimally projecting out the effect of the nuisance parameters $\theta_{N}$ and is the appropriate quantity entering the CR-type bound for $\theta_{I}$ [36,37].

Let $J_{\pi }$ be the classical Fisher information matrix of the prior π on $\theta_{N}$ of which the a,b-th entry is(10)$$\begin{array}{c} J_{\pi ,ab}:=\int_{\Theta_{N}}\left(\frac{\partial }{\partial \theta_{N,a}}\log\pi \left(\theta_{N}\right)\right)\left(\frac{\partial }{\partial \theta_{N,b}}\log\pi \left(\theta_{N}\right)\right)\pi \left(\theta_{N}\right)\;d\theta_{N}. \end{array}$$

We reserve the indices i,j for components of $\theta_{I}$ (written $\theta_{I,i},\theta_{I,j}$) and the indices a,b for components of $\theta_{N}$ (written $\theta_{N,a},\theta_{N,b}$). We define the *hpQFIM* by prior-averaging the blocks over π and taking the Schur complement:

**Definition** **2**(Hybrid partial quantum Fisher information matrix (hpQFIM))**.**(11)$$\begin{array}{c} J_{I|N}^{\left(\pi \right)}\left(\theta_{I}\right):=\;E_{\pi }\left[J_{II}\right]\;-\;E_{\pi }\left[J_{IN}\right]\;{\left(\;E_{\pi }\left[J_{NN}\right]+J_{\pi }\right)}^{-1}E_{\pi }\left[J_{NI}\right]. \end{array}$$

Intuitively, $J_{I|N}^{\left(\pi \right)}$ aggregates (i.e., prior-averages out) nuisance information and quantifies how much information for $\theta_{I}$ remains after accounting for the prior on $\theta_{N}$.

### 2.3. Hybrid CR-Type Lower Bound

We state the main result of the paper, which is proven by using the covariance inequality.

**Theorem** **1**(Quantum hybrid CR-type lower bound)**.** *For any POVM Π and any locally unbiased estimator for the parameters of interest ${\hat{\theta }}_{I}$, the matrix inequality*(12)$$\begin{array}{c} V_{\theta_{I},\pi }\left(\Pi ,{\hat{\theta }}_{I}\right)\;⪰\;{\left(J_{I|N}^{\left(\pi \right)}\left(\theta_{I}\right)\right)}^{-1}, \end{array}$$*holds. Hence, for any weight matrix W≻0, the hybrid risk is bounded as*
(13)$$\begin{array}{c} R_{\theta_{I},\pi }\left(\Pi ,{\hat{\theta }}_{I}|W\right)\;\geq \;\mathrm{Tr}\left[\;W\;{\left(J_{I|N}^{\left(\pi \right)}\left(\theta_{I}\right)\right)}^{-1}\right]. \end{array}$$

**Proof.** Details in Appendix A. □

**Remark** **2.**
*Finite-sample bias and benchmark choice.*


In finite-sample regimes, practically used estimators for $\theta_{I}$ (e.g., small-sample MLEs) can be biased with respect to the frequentist target. In such cases the locally-unbiased hybrid CRB provides an ideal benchmark but may not tightly track the empirical MSE. A practical rule of thumb is that, if one does use a biased estimator with bias vector $b(\theta_{I})$ and the derivative of bias $B\left(\theta_{I}\right)=\partial_{\theta_{I}}b\left(\theta_{I}\right)$, a modified information can be written as$$J_{h}^{\mathrm{mod}}\;=\;({\left(I+B\right)}^{⊤}{)}^{-1}\;J_{I|N}^{\left(\pi \right)}\left(\theta_{I}\right)\;{\left(I+B\right)}^{-1}.$$

A systematic derivation is left to future work (cf. general Cramér–Rao inequality for biased estimator in [38]).

Motivation for two-sided approximations.

The hpQFIM $J_{I|N}^{\left(\pi \right)}$ is the central information quantity in our framework, but it is not always the most convenient object to evaluate or compare across models and priors. In quantum point estimation with nuisance parameters, related results bound the Schur complement-type information between an averaging-after-inverse quantity and a simpler interest-block average. Such bounds serve two purposes: (i) they provide computationally convenient approximations when the exact partial information is hard to obtain, and (ii) they characterize how much precision can be gained or lost due to nuisance parameters and prior uncertainty. Motivated by this practice, we establish analogous two-sided approximations for the hybrid setting.

**Theorem** **2**(Lower and upper approximations for the hpQFIM)**.** *For the hpQFIM $J_{I|N}^{\left(\pi \right)}$, the following inequalities hold:*(14)$$\begin{array}{c} E_{\pi }\left[J_{II}\left(\theta_{I},\theta_{N}\right)\right]\;⪰\;J_{I|N}^{\left(\pi \right)}\left(\theta_{I}\right)\;⪰\;E_{\pi }\left[J_{I|N}\left(\theta_{I},\theta_{N}\right)\right]. \end{array}$$

**Proof.** Details in Appendix B. □

We have two remarks on the theorem.

Computational surrogates. For our model, the partial information $J_{I|N}$ has a closed form at each nuisance sample, so the right bound $E_{\pi }\left[J_{I|N}\left(\theta_{I},\theta_{N}\right)\right]$ is evaluated by averaging closed-form matrices and then inverting once. The left bound $E_{\pi }\left[J_{II}\left(\theta_{I},\theta_{N}\right)\right]$ is even simpler: average once and invert once.In contrast, the hybrid quantity requires the term ${\left(E_{\pi }\left[J_{NN}\right]+J_{\pi }\right)}^{-1}$, which does not admit a closed form in general (it depends on the prior π). Consequently, it typically has to be computed numerically and repeatedly (e.g., across prior hyperparameters), making this middle term the computational bottleneck. This is precisely why the left/right bounds are useful: they bracket $J_{I|N}^{\left(\pi \right)}$ while avoiding repeated inner inversions.

## 3. Examples (Noisy Qubit Metrology)

This section complements the hybrid framework in Section 2 by reporting numerical comparisons on qubit models in Bloch sphere parameters. An analytically solvable model will be presented in the next section.

### 3.1. Numerical Comparison on Qubit Models

We consider single-qubit models with two parameters $\theta =(\theta_{I},\theta_{N})$ and priors π on the nuisance. For each model we report (i) the lower bound for hybrid risk $\mathrm{Tr}\left[W\;{\left(J_{I|N}^{\left(\pi \right)}\left(\theta_{I}\right)\right)}^{-1}\right]$ (Definition 2) and (ii) the lower bound and upper bound for the hpQFIM (Theorem 2). Unless stated otherwise, we use W=I.

The ideal state in the Bloch vector representation is(15)$$\begin{array}{cc} \rho_{\theta_{I}} & =\frac{1}{2}\left(I+s_{\mathrm{ideal}}\left(\theta_{I}|r,\phi \right)\cdot \sigma \right), \end{array}$$(16)$$\begin{array}{cc} s_{\mathrm{ideal}}\left(\theta_{I}|r,\phi \right) & =(r\sin\phi \cos\theta_{I},r\sin\phi \sin\theta_{I},r\cos\phi ), \end{array}$$
where $\sigma =(\sigma_{x},\sigma_{y},\sigma_{z})$ is the vector of Pauli matrices and r,ϕ are fixed. Here, $\theta_{I}$ is the parameter to be estimated. Suppose this ideal state is affected by some unknown noise, which is characterized by a nuisance parameter $\theta_{N}$. Then the noisy model can be written in Bloch vector parameters as(17)$$\begin{array}{c} \rho_{\theta_{I},\theta_{N}}=\frac{1}{2}\left(I+s(\theta_{I},\theta_{N}|r,\phi )\cdot \sigma \right). \end{array}$$

In what follows, we consider three noisy models.

Phase with extra rotation: interest $\left(\theta_{I}\right)$, nuisance $\left(\theta_{N}\right)$; we sweep prior concentration to illustrate the predicted gap.(18)$$s\left(\theta_{I},\theta_{N}|r,\phi \right)=\left(r\sin\phi \cos\left(\theta_{I}+\theta_{N}\right),r\sin\phi \sin\left(\theta_{I}+\theta_{N}\right),r\cos\phi \right),$$
with parameters $\theta_{I}\in \left[0,2\pi \right),\;\theta_{N}\in \left[0,2\pi \right)$ and fixed values r∈(0,1),ϕ∈[0,2π). This model is impossible to estimate in point estimation but is tractable in this hybrid framework. The result is illustrated in Figure 1.Additional-sine model: interest $\left(\theta_{I}\right)$, nuisance $\left(\theta_{N}\right)$ with cross-coupling; priors on $\theta_{N}$ with varying concentration.(19)$$s\left(\theta_{I},\theta_{N}|r,\phi \right)=\left(r\sin\phi \cos\theta_{I},r\sin\phi \sin\left(\theta_{I}+\theta_{N}\right),r\cos\phi \right),$$
with parameters $\theta_{I}\in \left[0,2\pi \right),\;\theta_{N}\in \left[0,2\pi \right)$ and fixed values r∈(0,1),ϕ∈[0,2π) with a constraint $r^{2}\left(\sin^{2}\phi +1\right)\leq 1$. This model can be interpreted as a toy abstraction of situations where an additional phase affects only one channel (e.g., a single arm of an interferometer or one polarization component), thereby breaking the usual rotational symmetry. The result is illustrated in Figure 2.Anisotropic shrinking: dissipative channel with axis-dependent contraction; we isolate the interest while averaging nuisance shrinkage.(20)$$s\left(\theta_{I},\theta_{N}|r,\phi \right)=\left(r\sin\phi \cos\theta_{I},r\;\theta_{N}\sin\phi \sin\theta_{I},r\cos\phi \right),$$
with parameters $\theta_{I}\in \left[0,2\pi \right),\;\theta_{N}\in \left(0,1\right]$ and fixed values r∈(0,1),ϕ∈[0,2π). The result is illustrated in Figure 3.

Steps for numerical analysis.

We repeat the following steps to analyze each model. The results will be given in the next subsections.

Compute the QFIM $J(\theta_{I},\theta_{N})$ and the partial QFIM $J_{I|N}\left(\theta_{I},\theta_{N}\right)$ via Equation (Equation 9).Let π be the uniform distribution in the domain of the nuisance parameter (non-informative prior), and compute the analytical form of the hpQFIM (Definition 2) and its corresponding lower and upper approximations (Theorem 2.) One may notice that in Theorem 1, the lower bound of hybrid risk is the inverse of the hpQFIM. Thus, in this section, we demonstrate the lower and upper approximations in the inverse form as(21)$$\begin{array}{c} {\left(E_{\pi }\left[J_{I|N}\left(\theta_{I},\theta_{N}\right)\right]\right)}^{-1}\;⪰\;{\left(J_{I|N}^{\left(\pi \right)}\left(\theta_{I}\right)\right)}^{-1}\;⪰\;{\left(E_{\pi }\left[J_{II}\left(\theta_{I},\theta_{N}\right)\right]\right)}^{-1}. \end{array}$$Derive the values for the former quantities for grid points (fifty points in each figure) of the parameter of interest in its range and truncate the point on the boundary.

### 3.2. Extra Rotation Model

For the model$$s\left(\theta_{I},\theta_{N}|r,\phi \right)=\left(r\sin\phi \cos\left(\theta_{I}+\theta_{N}\right),\;r\sin\phi \sin\left(\theta_{I}+\theta_{N}\right),\;r\cos\phi \right),$$
the score directions satisfy $\partial_{\theta_{I}}s=\partial_{\theta_{N}}s$. Hence the single-qubit QFIM blocks are constants,$$J_{II}=J_{NN}=J_{IN}=r^{2}\sin^{2}\phi ,$$
independent of the parameters $\left(\theta_{I},\theta_{N}\right)$, and the likelihood-only Schur complement is identically zero: $J_{I|N}=J_{II}-J_{IN}^{2}/J_{NN}=0$. The result is shown in Figure 1.

We now discuss consequences seen in the plots. To simplify the notation, denote the three quantities with the following equations:$$U:={\left(E[J_{I|N}]\right)}^{-1},\;M:={\left(J_{I|N}^{\left(\pi \right)}\right)}^{-1},\;L:={\left(E[J_{II}]\right)}^{-1}.$$The four panels (for $\left(r,\phi \right)\in \{\left(0.5,\frac{\pi }{2}\right),\;\left(0.5,\frac{\pi }{3}\right),\;\left(0.3,\frac{\pi }{2}\right),\;\left(0.7,\frac{\pi }{2}\right)\}$) in Figure 1 exhibit the following:**Flat (constant) curves in $\theta_{I}$.** Since $J_{II},J_{NN},J_{IN}$ do not depend on angles, both *M* and *L* are constant in $\theta_{I}$. This matches the horizontal lines in all panels.**Divergent upper bound.** Because $J_{I|N}\equiv 0$, we have $E[J_{I|N}]=0$ and therefore$$U={\left(E[J_{I|N}]\right)}^{-1}=+\infty ,$$
as indicated by the “U=+∞” annotation.**Lower and middle terms.** Averaging yields$$L=\frac{1}{E[J_{II}]}=\frac{1}{r^{2}\sin^{2}\phi },\;J_{I|N}^{\left(\pi \right)}=\frac{r^{2}\sin^{2}\phi \;J_{\pi }}{r^{2}\sin^{2}\phi +J_{\pi }}\;⟹\;M=\frac{r^{2}\sin^{2}\phi +J_{\pi }}{r^{2}\sin^{2}\phi \;J_{\pi }},$$
where $J_{\pi }>0$ denotes the prior Fisher information for the nuisance. Thus M>L and the hybrid inequality U≥M≥L holds everywhere.

Next, we consider scaling with (r,ϕ). From the analytical expressions, we immediately see that only the scale $r^{2}\sin^{2}\phi$ matters:$$L=\frac{1}{r^{2}\sin^{2}\phi }\;\mathrm{and}\;M=\frac{r^{2}\sin^{2}\phi +J_{\pi }}{r^{2}\sin^{2}\phi \;J_{\pi }}.$$Therefore increasing *r* (with ϕ fixed) or increasing sinϕ (with *r* fixed) uniformly lowers both constant lines. This is exactly what is observed when comparing r=0.3 vs. 0.7 at ϕ=π/2, and ϕ=π/2 vs. π/3 at r=0.5.

To summarize the result of this model, we conclude as follows. The extra-rotation coupling makes the score directions for $\theta_{I}$ and $\theta_{N}$ collinear, so the likelihood-only Schur complement $J_{I|N}$ vanishes identically and pure likelihood information on $\theta_{I}$ is lost. Introducing a nonzero prior concentration $J_{\pi }$ on the nuisance regularizes this degeneracy: the hybrid information becomes $J_{I|N}^{\left(\pi \right)}=r^{2}\sin^{2}\phi J_{\pi }/\left(r^{2}\sin^{2}\phi +J_{\pi }\right)$, yielding the finite middle curve $M={\left(J_{I|N}^{\left(\pi \right)}\right)}^{-1}=\left(r^{2}\sin^{2}\phi +J_{\pi }\right)/\left(r^{2}\sin^{2}\phi J_{\pi }\right)$, while the lower bound $L={\left(E\left[J_{II}\right]\right)}^{-1}=1/r^{2}\sin^{2}\phi$ represents the naive baseline. The observed flatness of *M* and *L* in $\theta_{I}$ reflects that, for this model, only the global scale $r^{2}\sin^{2}\phi$ and the prior concentration $J_{\pi }$ determine estimation precision.

### 3.3. Additional Sine Model

For the model$$s\left(\theta_{I},\theta_{N}|r,\phi \right)=\left(r\sin\phi \cos\left(\theta_{I}\right),\;r\sin\phi \sin\left(\theta_{I}+\theta_{N}\right),\;r\cos\phi \right),$$
with interest $\theta_{I}$ and nuisance $\theta_{N}∼\mathrm{Unif}\left(0,1\right]$. The result is shown in Figure 2.

We discuss consequences of the plots. Denote, as before,$$U:={\left(E_{\pi }\left[J_{I|N}\right]\right)}^{-1},\;M:={\left(J_{I|N}^{\left(\pi \right)}\right)}^{-1},\;L:={\left(E_{\pi }\left[J_{II}\right]\right)}^{-1}.$$

First, we see that across all parameter choices, the ordering U≥M≥L holds for every $\theta_{I}$. The curves are symmetric about $\theta_{I}=\pi /2$ and reach their minima near the center of the interval. As $\theta_{I}\to 0$ or π, *M* and *L* grow rapidly, reflecting the near-collinearity of the score directions for $\theta_{I}$ and $\theta_{N}$ at the endpoints. In the bulk of the interval, *U* and *M* are nearly indistinguishable under the uniform prior, showing that averaging largely cancels cross-term fluctuations.

Next, we analyze the dependence on the model parameters (r,ϕ). The overall information scale $r^{2}\sin^{2}\phi$ governs the depth and position of the curves. Increasing *r* at fixed ϕ uniformly lowers all bounds and makes the central valley deeper. Reducing sinϕ at fixed *r* (e.g., $\phi :\;\frac{\pi }{2}\to \frac{\pi }{3}$) decreases $r^{2}\sin^{2}\phi$ and shifts the curves upward while preserving their shapes. These monotone trends are consistently observed across the four chosen parameter settings.

We hence summarize the second model as follows. The additional sine model highlights how an asymmetric coupling of the nuisance to the signal modifies estimation: the nuisance phase enters only the second transverse component, so the two parameter directions do not affect the state in a rotationally symmetric way. With a uniform prior over the nuisance, the lower bound $L={\left(E_{\pi }\left[J_{II}\right]\right)}^{-1}$ is a proxy in the bulk of $\theta_{I}$, but it becomes optimistic in a narrow neighborhood of the endpoints ($\theta_{I}\;\to \;0,\pi$), where the middle curve $M={\left(J_{I|N}^{\left(\pi \right)}\right)}^{-1}$ better reflects the true loss of information. This behavior follows from the anisotropic suppression of the Schur complement: as the score vectors $\partial_{\theta_{I}}s$ and $\partial_{\theta_{N}}s$ become nearly collinear near the endpoints, the conditional information $J_{I|N}$ is reduced more strongly than $J_{II}$.

### 3.4. Anisotropic Shrinking Model

We consider the anisotropic shrinking model$$s\left(\theta_{I},\theta_{N}|r,\phi \right)=\left(r\sin\phi \cos\theta_{I},\;r\;\theta_{N}\sin\phi \sin\theta_{I},\;r\cos\phi \right),$$
with interest $\theta_{I}$ and nuisance $\theta_{N}∼\mathrm{Unif}\left(0,1\right]$. The result is shown in Figure 3.

We discuss the properties of this model. Denote, as before,$$U:={\left(E_{\pi }\left[J_{I|N}\right]\right)}^{-1},\;M:={\left(J_{I|N}^{\left(\pi \right)}\right)}^{-1},\;L:={\left(E_{\pi }\left[J_{II}\right]\right)}^{-1}.$$

First, we look at ordering and endpoint behavior. Across all panels the hybrid inequality U≥M≥L holds pointwise in $\theta_{I}$. With a uniform prior over $\theta_{N}$, the plots are approximately symmetric about $\theta_{I}=\pi /2$ and attain their minima near the center. This symmetry follows from the trigonometric structure of the model, in which only $\sin\theta_{I}$ and $\cos\theta_{I}$ enter the QFIM blocks. As $\theta_{I}\to 0$ or π, both *U* and *M* rise rapidly, while *L* increases only moderately. This reflects a geometric degeneracy at the endpoints: $\partial_{\theta_{I}}s$ and $\partial_{\theta_{N}}s$ both carry a factor $\sin\theta_{I}$, so their norms—and hence the Schur complement $J_{I|N}$—are strongly suppressed there. Averaging over $\theta_{N}$ therefore drives $E_{\pi }\left[J_{I|N}\right]$ to small values, inflating *U* (and, to a lesser extent, *M*), whereas $J_{II}$ alone does not vanish at the same rate, keeping *L* comparatively low. Thus *L* becomes loose in a narrow neighborhood of the endpoints, while *M* better reflects the cost of eliminating the nuisance.

Next, tightness in the central region should be stressed. Around the symmetric center $\theta_{I}\approx \pi /2$, the three curves approach one another and the gap U−M becomes small; *M* nearly coincides with *U* and also approaches *L*. Away from the degeneracy, “average of Schur” and “Schur of averages” yield very similar information.

Last, we examine the dependence on the model parameters (r,ϕ). The panels for$$\left(r,\phi \right)\in \{\left(0.5,\frac{\pi }{2}\right),\;\left(0.5,\frac{\pi }{3}\right),\;\left(0.3,\frac{\pi }{2}\right),\;\left(0.7,\frac{\pi }{2}\right)\}$$
exhibit the expected following two trends. First, increasing *r* at fixed ϕ (compare r=0.3 vs. 0.7 at ϕ=π/2) increases Fisher information and hence lowers all three curves uniformly; the valley near $\theta_{I}=\pi /2$ deepens. Second, reducing sinϕ at fixed *r* (e.g., $\phi :\;\frac{\pi }{2}\to \frac{\pi }{3}$ for r=0.5) weakens the transverse component and raises all curves, with shapes essentially unchanged.

We summarize the behaviors of this model as follows. The anisotropic contraction along the nuisance-controlled axis amplifies endpoint degeneracy and creates a clear separation between *M* and *L* only where $\sin\theta_{I}$ is small. Overall, the estimation precision is controlled chiefly by the scale rsinϕ and by avoiding the endpoint region, providing concrete guidance for operating points in hybrid estimation.

## 4. Example (Direction Estimation)

In this section we analyze a qubit model that admits closed-form expressions in the multiparameter setting, and then specialize it to a hybrid framework where a prior is placed only on the radial nuisance parameter *r*, while the directional parameters (θ,ϕ) are estimated in the frequentist sense. The benchmark is directly linked to entropic characteristics of the state: for a qubit with Bloch radius *r*, the spectrum is {(1±r)/2} and the von Neumann entropy S(ρ) is a monotone function of *r*, making radius estimation operationally relevant for purity assessment.

### 4.1. Example: Bloch-Radius Model with Directional Interest and Radial Nuisance

We consider single-qubit states in Bloch form:(22)$$\rho \left(r,\theta ,\phi \right)=\frac{1}{2}\left[I+r\left(\sin\theta \cos\phi \;\sigma_{x}+\sin\theta \sin\phi \;\sigma_{y}+\cos\theta \;\sigma_{z}\right)\right],$$
where the parameters of interest are the spherical angles (θ,ϕ)∈[0,π]×[0,2π) and the nuisance parameter is the Bloch radius r∈(0,1). Let $s\left(r,\theta ,\phi \right)=r\;\hat{n}\left(\theta ,\phi \right)$ with$$\hat{n}\left(\theta ,\phi \right)=\left(\sin\theta \cos\phi ,\;\sin\theta \sin\phi ,\;\cos\theta \right).$$The parameter derivatives of the Bloch vector are$$\partial_{r}s=\hat{n},\;\partial_{\theta }s=r\;\partial_{\theta }\hat{n},\;\partial_{\phi }s=r\;\partial_{\phi }\hat{n},$$
and the elementary spherical identities,$$\hat{n}\;\cdot \;\partial_{\theta }\hat{n}=\hat{n}\;\cdot \;\partial_{\phi }\hat{n}=0,\;∥\partial_{\theta }\hat{n}∥=1,\;∥\partial_{\phi }\hat{n}∥=\sin\theta ,\;\det\left[\partial_{\theta }\hat{n},\hat{n},\partial_{\phi }\hat{n}\right]=-\sin\theta ,$$
will be used below.

SLD quantum Fisher information.

The QFIM for the parameter order (r,θ,ϕ), i.e., $\theta_{1}=r,\theta_{2}=\theta ,\theta_{3}=\phi$, is$$J\left(r,\theta ,\phi \right)=\mathrm{diag}\;\left({\left(1-r^{2}\right)}^{-1},\;r^{2},\;r^{2}\sin^{2}\theta \right).$$

We thus have $J_{r\theta }=J_{r\phi }=J_{\theta \phi }=0$. Recall that the parameters of interest are $\theta_{I}=\left(\theta ,\phi \right)$, whereas the nuisance parameter is $\theta_{N}=r$.

Hybrid partial information and CR-type bound.

Treating (θ,ϕ) as interest and *r* as nuisance, the partial SLD information (conditioning on *r*) is the Schur complement:$$J_{I|N}=J_{II}-J_{IN}{\left(J_{NN}\right)}^{-1}J_{NI}.$$Since the cross-terms vanish, we have(23)$$J_{I|N}\left(\theta ,\phi ;\;r\right)=J_{II}\left(\theta ,\phi ;\;r\right)=r^{2}\left(\begin{array}{cc} 1 & 0\\ 0 & \sin^{2}\theta \end{array}\right).$$

Consequently, for any prior π on *r* the three quantities of Theorem 2 are identical, i.e.,(24)$$\begin{array}{c} {\left(E_{\pi }\left[J_{I|N}\left(\theta ,\phi \right)\right]\right)}^{-1}={\left(J_{I|N}^{\left(\pi \right)}\left(\theta ,\phi \right)\right)}^{-1}={\left(E_{\pi }\left[J_{II}\left(\theta ,\phi \right)\right]\right)}^{-1}=\frac{1}{E_{\pi }\left[r^{2}\right]}\left(\begin{array}{cc} 1 & 0\\ 0 & \sin^{-2}\theta \end{array}\right). \end{array}$$

Thus, in point estimation (oracle *r* known) the attainable variance for unbiased estimation of (θ,ϕ) depends on the value of *r* via (Equation 23); in the hybrid setting it depends on the prior π through $E_{\pi }\left[r^{2}\right]$. Therefore, one can certify a direction-estimation risk lower bound via the prior without knowing the true *r*.

### 4.2. Numerical Simulation: One-Parameter Qubit Gate

We consider the one-parameter qubit gate $U_{\phi }=e^{-i\phi \sigma_{z}/2}$ acting on $\rho_{0}=\frac{1}{2}\;\left(I+r\;\sigma_{x}\right)$, yielding a two-parameter model $\rho \;\left(\phi ,r\right)=\frac{1}{2}\;\left[I+r(\cos\phi \;\sigma_{x}+\sin\phi \;\sigma_{y})\right]$. Here ϕ is the parameter of interest and *r* is the nuisance parameter. This is an extension of the one-parameter qubit gate model [39]. Measurements are ordinary PVMs along *x* and *y* (settings ω∈{0,π/2}); a total sample size *M* is split equally across the two measurement settings. The nuisance parameter purity r∈(0,1) is drawn once per dataset from Beta distribution π(r)=Beta(a,b) (nuisance; we use a=10, b=2 as an example) and marginalized in the likelihood.

#### 4.2.1. Estimator

For a dataset $m={\left\{\left(M_{\omega },m_{\omega }^{+}\right)\right\}}_{\omega }$,$$L\left(\phi |m\right)\;=\;\int_{0}^{1}\prod_{\omega \in \{0,\pi /2\}}{\left[p_{\omega }\left(+\;|\;\phi ,r\right)\right]}^{m_{\omega }^{+}}\;{\left[1-p_{\omega }\left(+\;|\;\phi ,r\right)\right]}^{M_{\omega }-m_{\omega }^{+}}\;\pi \left(r\right)\;dr,$$
with $p_{\omega }\left(+\;|\;\phi ,r\right)=\frac{1+r\;\cos(\phi -\omega )}{2}$. We take ${\hat{\phi }}_{\mathrm{MLE}}=\arg\max_{\phi }L\left(\phi |m\right)$. To perform this method we use Gauss–Legendre quadrature for the integral and dense grid plus local quadratic refinement for the maximization.

#### 4.2.2. Metric and hpQFIM Baseline

We report the error MSE, where $\mathrm{MSE}=E[{\left(\hat{\phi }-\phi^{*}\right)}^{2}]$. By the hpQFIM in Theorem 1, for the split (ϕ,r) one has $J_{I|N}^{\left(\pi \right)}\left(\phi \right)=E_{\pi }\left[r^{2}\right]$ (Equation (Equation 24)), hence the hybrid CR-type bound is$$\mathrm{MSE}\;\geq \;\frac{1}{M\;E_{\pi }\left[r^{2}\right]}.$$

For π=Beta(a,b), $E_{\pi }\left[r^{2}\right]=\frac{a(a+1)}{\left(a+b)(a+b+1\right)}$, so the baseline appears as a curve $y=1/(M\times E_{\pi }\left[r^{2}\right])$ in our plots.

#### 4.2.3. Visualization

We fix $\phi^{*}=1.1$ (rad), take *M* on 30 equally spaced values from 100 to 1000, and run T=1000 independent replicates per *M*. For each replicate, draw r∼Beta(10,2), simulate binomial counts under the two PVM settings, compute ${\hat{\phi }}_{\mathrm{MLE}}$, and record ${\left(\hat{\phi }-\phi^{*}\right)}^{2}$. For each *M* we display a boxplot with the mean line in green; we overlay the hpQFIM baseline $y=1/(M\;E_{\pi }\left[r^{2}\right])$ in blue dashed line. The results are plotted in Figure 4.

We remark that with fixed x/y PVMs and dataset-level nuisance, the mean value of MSE always sits above the hpQFIM (quantum bound), showing the validity of our lower bound.

## 5. Conclusions

In this paper, we proposed a hybrid estimation framework that treats parameters of interest and nuisance parameters asymmetrically by placing a prior only on the nuisance sector while estimating the parameter of interest in the frequentist sense. Within this framework, we (i) defined the hybrid partial quantum Fisher information matrix (hpQFIM) by prior-averaging the nuisance block and taking the Schur complement on the interest block; (ii) derived the associated hybrid Cramér–Rao-type (CR-type) bound; (iii) clarified the operational gain over pure point estimation—hybrid-optimal measurements depend on the prior over the nuisance rather than its unknown true value; and (iv) established inequalities that relate prior-averaged quantities and elucidate their limiting behaviors.

To make the discussion concrete, we analyzed an analytically solvable single-qubit model where the direction (θ,ϕ) is the parameter of interest and the Bloch radius *r* plays the role of nuisance (with a prior π). Because the QFIM is diagonal (cross-terms vanish), the hybrid partial information for (θ,ϕ) reduces to the prior average of the interest block, yielding the CR-type matrix bound$${\left(J_{I|N}^{\left(\pi \right)}\right)}^{-1}=\frac{1}{E_{\pi }\left[r^{2}\right]}\;\mathrm{diag}\;\left(1,\sin^{-2}\theta \right),$$
while with oracle knowledge of *r* the bound scales as $1/r^{2}$; under a genuine prior, the hybrid quantities coincide.

Beyond this solvable case, our inequalities position the hybrid bound between natural Bayesian and point-estimation surrogates, thereby quantifying how prior knowledge about nuisance parameters can be systematically leveraged without fully committing to a Bayesian treatment of all parameters. These results give a unified and operationally transparent picture of nuisance handling in quantum metrology.

Several directions follow naturally. We list three possible extensions of this work.

Tightness and achievability. Determining conditions under which the hybrid CR-type bound is tight and characterizing the structure of achieving measurements—especially beyond the qubit radius example—remain open. Connections to D-invariant models and to measurement classes with symmetry constraints are promising [40].Prior modeling and robustness. Moving from uniform priors to anisotropic families such as von Mises–Fisher priors enables a controlled interpolation between ignorance and alignment. Quantifying robustness of hybrid-optimal measurements against prior misspecification is an important practical question.Full hybrid model and bias-aware hpQFIM. Beyond the canonical hybrid setting (random nuisances; pointwise interest), a natural next step is a full hybrid framework in which parameters are partitioned into four classes: interest–random, interest–nonrandom, nuisance–random, and nuisance–nonrandom. Within this program we will also develop a bias-aware extension of our hpQFIM guidance for finite-sample practice [41]: for biased estimators of the interest–random block, a modified hybrid information with the bias provides the natural counterpart of our CR-type results.

Overall, the hybrid viewpoint separates what must be learned (the parameter of interest) from what can be integrated out using prior structure (the nuisance), yielding bounds and design principles that are both rigorous and operationally meaningful. We expect this perspective to be broadly useful for quantum metrology in low-copy and resource-constrained scenarios where nuisance is inevitable.

## Figures and Tables

**Figure 1 entropy-27-01184-f001:**
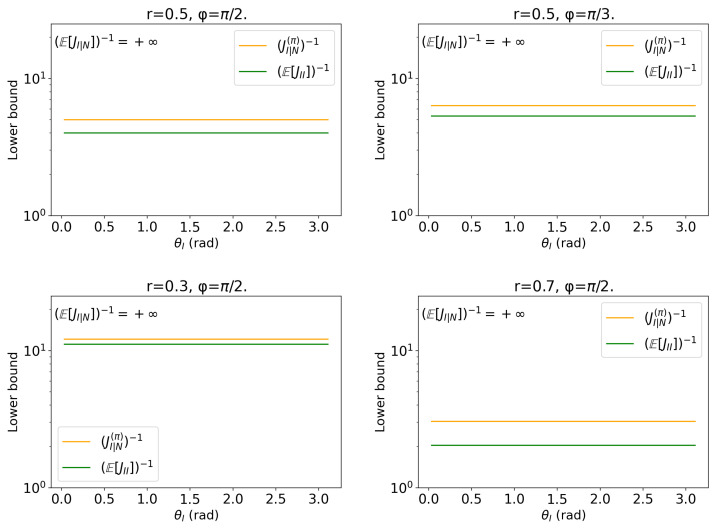
Comparison of the proposed bound based on hpQFIM (Theorem 1) and its two approximations (Equation (Equation 21)) for the model (phase with extra rotation). The model parameters are set as (r=0.5,ϕ=π/2), (r=0.5,ϕ=π/3), (r=0.3,ϕ=π/2), and (r=0.7,ϕ=π/2) with uniform $\theta_{N}∼U\left[0,2\pi \right)$. The bound ${\left(J_{I|N}^{\left(\pi \right)}\right)}^{-1}$ is given in orange. The lower approximation ${\left(E\left[J_{II}\right]\right)}^{-1}$ is in green. These bounds are flat because of the unitary model. The upper approximation is infinite due to the fact that the score directions are collinear.

**Figure 2 entropy-27-01184-f002:**
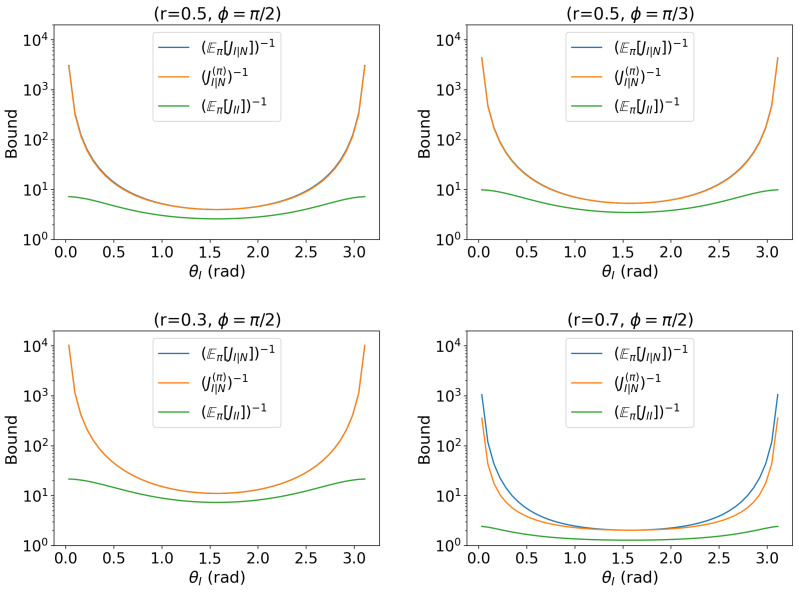
Comparison of the proposed bound based on hpQFIM (Theorem 1) and its two approximations (Equation (Equation 21)) for the model (additional sine rotation). The model parameters are set as (r=0.5,ϕ=π/2), (r=0.5,ϕ=π/3), (r=0.3,ϕ=π/2), and (r=0.7,ϕ=π/2) with uniform $\theta_{N}∼U\left[0,2\pi \right)$. The bound ${\left(J_{I|N}^{\left(\pi \right)}\right)}^{-1}$ is given in orange. The lower approximation ${\left(E\left[J_{II}\right]\right)}^{-1}$ is in green. The upper approximation ${\left[E\left(J_{I|N}\right)\right]}^{-1}$ is in blue. The endpoints of blue and orange curves represent degeneracy since the score directions are nearly collinear.

**Figure 3 entropy-27-01184-f003:**
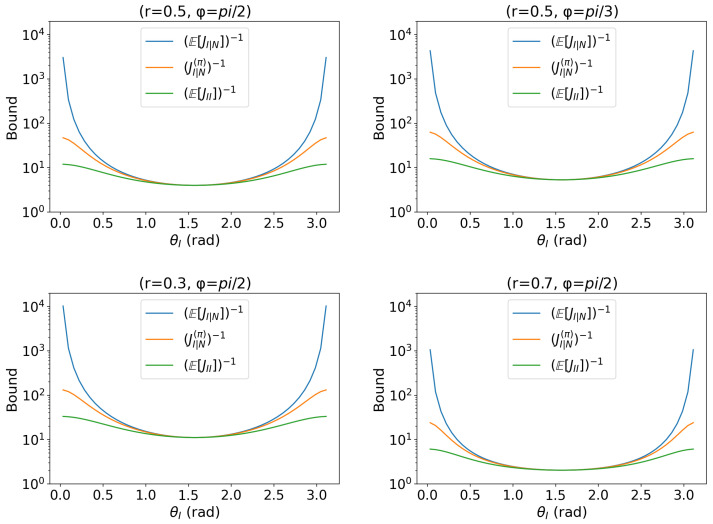
Comparison of the proposed bound based on hpQFIM (Theorem 1) and its two approximations (Equation (Equation 21)) for the model (anisotropic shrinking). The model parameters are set as (r=0.5,ϕ=π/2), (r=0.5,ϕ=π/3), (r=0.3,ϕ=π/2) and (r=0.7,ϕ=π/2) with uniform $\theta_{N}∼U\left[0,1\right]$. The bound ${\left(J_{I|N}^{\left(\pi \right)}\right)}^{-1}$ is given in orange. The lower approximation ${\left(E\left[J_{II}\right]\right)}^{-1}$ is in green. The upper approximation ${\left[E\left(J_{I|N}\right)\right]}^{-1}$ is in blue. The endpoints of blue curves represent degeneracy since the score directions are nearly collinear.

**Figure 4 entropy-27-01184-f004:**
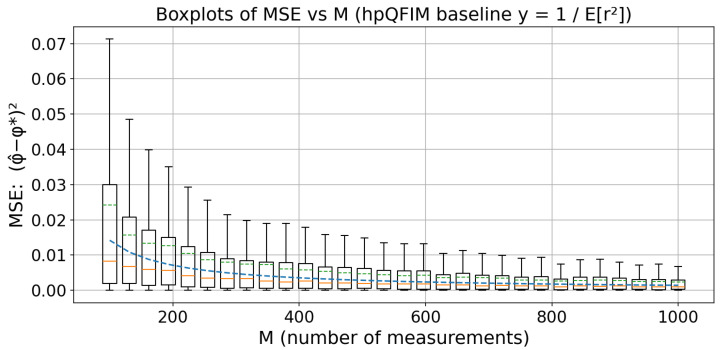
Boxplots of MSE versus *M* for the one-parameter gate (PVM angular ω∈{0,π/2}, nuisance parameter r∼Beta(10,2), number of replicates T=1000 per *M*). The dashed green line is the mean and the orange line is the median. The dashed blue line is the hpQFIM baseline $y=1/(M\;E_{\pi }\left[r^{2}\right])$.

## Data Availability

The data and program code that support the findings of this study are available from the corresponding author upon reasonable request.

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
