# Peer review of "Hybrid Cramér-Rao Bound for Quantum Bayes Point Estimation with Nuisance Parameters"

_entropy, 2025, doi:10.3390/e27121184_

Round 1
Reviewer 1 Report
Comments and Suggestions for Authors
The manuscript addresses a hybrid estimation framework in which parameters of interest are treated in the pointwise (frequentist) sense while nuisance parameters are integrated out under a prior. The central technical object is the hybrid partial QFIM (hpQFIM), a prior-averaged Schur complement—which yields a CR-type lower bound on the hybrid risk (Theorem 1), and is bracketed by two computable surrogates (Theorem 2). The paper gives qubit examples (including analytically solvable direction estimation with radial nuisance) and numerics that illustrate how prior knowledge over nuisances regularizes otherwise ill-posed point estimation. (See Definition 3, Eq. (16); Theorem 1, Eq. (17); Theorem 2, Eq. (19); the qubit example Eqs. (26)–(27); and the plots in Figures 1–3, e.g., Figure 1 on page 8 shows the flat curves when the score directions are colinear.)
The paper is clearly written, technically sound, and timely. I suggest one focused addition (bias-aware, non-asymptotic analysis for Bayes estimators of the interest parameters) plus a few clarifications and minor edits.
Add a short subsection: Bias of Bayes estimators and a modified Fisher information in the non-asymptotic regime. The motivation for this is the following: In finite-copy regimes, Bayes-optimal estimators for the parameters of interest (e.g., posterior means) are typically biased relative to the frequentist target. The authors current framework enforces local unbiasedness for (Eq. (3)) and derives an ideal CR-type matrix bound; however, practitioners using truly Bayes-optimal estimators will want guidance that remains valid without the local-unbiasedness constraint.
I would also suggeste experimental illustration, for examples:one-parameter qubit gate.
- A clear demonstration would significantly strengthen Section 3 in the manuscript.
- Please highlight earlier (end of Sec 2.1) that the main theorems assume local unbiasedness in and a -independent prior over ; the boundary-term conditions used in Appendix A could be restated in the main text.
- Figure 1 (p. 8) nicely shows the constant curves in the extra-rotation model; adding a small inset explaining “colinear score directions ⇒ ” would help new readers. Likewise, calling out the endpoint degeneracies in Figures 2–3 in their captions would align with the text on pages 9–10.
- A few typos issue and some spacing around equations) can be cleaned at copy-edit.
The paper is strong and publishable after major revision. The bias-aware, non-asymptotic subsection and a compact experimental qubit-gate illustration would broaden its practical impact and connect your hpQFIM bound to what experimentalists actually estimate in finite-data settings.
Reviewer 2 Report
Comments and Suggestions for Authors
The authors proposed a hybrid estimation framework, such as hybrid partial quantum Fisher information matrix and hybrid Quantum Cramer-Rao-type bound. The idea is quite interesting. However it is lack of a presentation skill. Specifically, it is hard to understand the examples with the figures, where the rotation model, the additional-sine model, and the anistropic shrinking model are distracted. Please make it clear with revision.
Comments on the Quality of English LanguagePlease check the grammar and typos.
Round 2
Reviewer 1 Report
Comments and Suggestions for Authors
Dear Authors,
Thank you for the thorough revision. The new version addresses all of my earlier concerns. The manuscript is clearly written, technically sound, and timely.
I appreciate the additions and clarifications, including the treatment of bias in finite-sample Bayes estimation, the tightened discussion of assumptions (local unbiasedness for the interest parameters). These changes make the results much more accessible to experimental readers. I now recommend the manuscript for publication.